# Health Care Accessibility and Breast Cancer Mortality in Europe

**DOI:** 10.3390/ijerph192013605

**Published:** 2022-10-20

**Authors:** Agata Ciuba, Katarzyna Wnuk, Aneta Nitsch-Osuch, Marta Kulpa

**Affiliations:** 1Department of Social Medicine and Public Health, Doctoral School, Medical University of Warsaw, 02-007 Warsaw, Poland; 2Maria Sklodowska-Curie Research Institute of Oncology, 02-034 Warsaw, Poland; 3Department of Health Policy Programs, Department of Health Technology Assessment, Agency for Health Technology Assessment and Tariff System, 00-032 Warsaw, Poland; 4Department of Social Medicine and Public Health, Medical University of Warsaw, 02-007 Warsaw, Poland; 5Department of Psychology and Medical Communication, Medical University of Warsaw, 00-581 Warsaw, Poland

**Keywords:** breast cancer mortality, preventive mammography, health care accessibility

## Abstract

Background: Breast cancer is the most common cause of death, due to malignant neoplasms in women worldwide. The nature of the symptoms associated with breast cancer depends on the stage of the disease. In this case, any cancerous changes in the initial phase of the disease can only be detected during imaging tests. Participation in mammography screening can reduce breast cancer mortality by up to 40%, if only 70% of the eligible population participates in preventive programs. The purpose of the study was to assess the impact of accessibility to health care resources on breast cancer mortality. Methods: Eurostat aggregated health care data was extracted. Hierarchical cluster analysis of average breast cancer mortality identified four groups of countries with similar mortality rates and trends. The data was then analyzed, in terms of access to health care. Results: It was observed that the higher the financial expenditure on health care and the better the health care accessibility, the lower the mortality rates of breast cancer. Conclusions: There are examples indicating that the studied elements are not the only factors affecting the improvement of population health. The authors would like to emphasize the need to influence lifestyle factors, direct cancer risk, and introduce a multidisciplinary approach to breast cancer prevention.

## 1. Introduction

Presented by Marc Lalonde in the 1970s, the Health Field Concept says that four factors are responsible for health—human biology, environment, lifestyle, and health care organization. The human biology element is limited by biological and genetic conditions—these are factors that can cause hereditary predisposition to certain diseases or health problems. The environment category is related to the body’s external or physical environment, including the social community. The lifestyle category refers to individual’s acquired daily habits, behaviors, and activities undertaken. The final category of the health field concept is the health care organization, which consists of people, hospital infrastructure, the quality and quantity of services provided, out-of-hospital care, emergency rooms, and other health services [1]. 

Breast cancer is the most commonly occurring cancer and the leading cause of death, due to malignant neoplasm, in women worldwide. It translates into 24.5% of all cancer diagnoses and 15.5% of all cancer deaths [2]. Similarly, in European countries, breast cancer is diagnosed among 28.7% [3] of cancer female patients and contributes to 16.5% of cancer deaths [4]. 

The etiology of breast cancer is not fully known, but there are several identified factors that increase the risk of the disease, such as genetic burden (presence of BRCA1/2 gene mutations), age (80% of cases occur after the age of 50), history of prolonged exposure to sex hormones (i.e., early menarche, late menopause), years of hormone replacement therapy, lifestyle (poor diet and obesity, insufficient physical activity, frequent alcohol consumption), and history of previous breast diseases [5]. 

The nature of the symptoms associated with breast cancer depends on the stage of the disease. The most common symptom of breast cancer is a malignant tumor (hard, with an uneven surface, and well-defined borders), skin and nipple lesions, and enlargement of the surrounding lymph nodes. Pain rarely accompanies the primary breast lesion. More often it is the result of advanced diseases, such as distant metastasis. In the case of this cancer, the initial phase of the disease is asymptomatic. Any cancerous changes can only be detected during preventive mammography or other imaging tests. In contrast, the first physical symptoms often appear only at an advanced stage of the disease [6]. 

Imaging tests can detect cancerous lesions, determine the stage, evaluate treatment results, and monitor the course of the disease. In the diagnosis of breast diseases, tests such as X-ray mammography (MMG), ultrasonography (USG), magnetic resonance (MR), and positron emission tomography (PET) are used [7].

MMG and USG are the primary methods of diagnostic breast imaging. Mammography is used in cases of clinical suspicion of breast disease, in particular, in screening, due to its high specificity and sensitivity. On the basis of MMG, a “radiologically suspicious neoplastic lesion” can be distinguished. The nature of the lesion can only be determined by cytological or histological verification. It is generally not used in women under the age of 35 [7]. 

USG is an examination with higher sensitivity and lower specificity, compared to MMG. However, it is very useful in differentiating solid and cystic lesions [8,9]. In addition, ultrasound is recommended in the younger age group, due to the dense and glandular breast structure typical of women under 40 [7]. MR has high sensitivity and specificity. It is used in the search for the primary outbreak in patients with lymph node metastasis and in diagnosing breast lesions in BRCA gene mutation carriers. Structural, as well as functional, assessments of tissues can be made on the basis of MR examination [7]. PET, due to its low sensitivity and inability to detect lesions smaller than 1 cm in diameter, is not used in the everyday diagnosis of breast cancer. Instead, it allows for the differentiation of malignant and benign lesions. Primarily, it is used to identify distant metastases. Similarly, the role of computer tomography in the initial diagnosis of breast cancer is limited, but it is important in the diagnosis of distant metastases [7]. 

The goal of imaging tests is to detect breast cancer at the earliest possible stage. Early detection and medical intervention greatly increase the treatment efficacy. Late diagnosis is the biggest challenge to the health care systems. Participation in mammography screening can reduce breast cancer mortality by up to 40% [10], due to the possibility of detecting precancerous conditions and cancer at the earliest possible stage. However, young age decreases the effectiveness of the screening. Undergoing mammography examination at the age of 20–30 reduces the risk of developing advanced stages of breast cancer by 10%, at age 40 by 20%, and at age 50 or 60 by 35% [11]. The effects of a 40% reduction in breast cancer mortality can only be achieved by mass participation of women in screening—a minimum of 70% of the population eligible for screening [12]. 

It needs to be highlighted that breast cancer mortality is an emerging and constantly growing global health problem that can be reduced through appropriate interventions. Previously published studies focused either on breast cancer treatment or the effectiveness of public health initiatives. This article shows an analysis of health care accessibility and breast cancer mortality in European countries with different economic levels. The conclusions that increased health care funding and better accessibility to health care are associated with lower breast cancer mortality rates should not come as a surprise. However, it is important to remember that adequate infrastructure and conditions conducive to recovery are not enough to prevent breast cancer mortality. Public health initiatives to prevent the disease and, consequently, reduce breast cancer mortality should be supported by effective communication.

The aim of the study was to analyze European data on breast cancer mortality from the years 2011–2017 and to assess the impact of accessibility to health care resources on this indicator, with particular emphasis on medical infrastructure and medical personnel. 

## 2. Materials and Methods

The study was based on aggregated health care data. The presented data was acquired from the European Statistical Office (Eurostat). The data was analyzed in terms of accessibility to health care, regarding the following factors: health care expenditure; the proportion of women that underwent a mammogram in the past two years; health care availability per capita: number of practicing physicians, nurses, and midwives, number of magnetic resonance imaging units and computer tomography scanners, and number of available hospital beds. 

Mortality trends for European countries were categorized into groups of maximum similarity between countries, in terms of breast cancer mortality rate. Its changes over time were analyzed using the hierarchical cluster analysis method with vector Pearson correlation as a measurement to identify groups of countries with similar mortality rates and trends. The average normalized metrics of health care quality and accessibility indicators were compared in the groups of countries described above.

The analysis includes data for the period between the years 2011–2017. Therefore, the time of the COVID-19 pandemic and its undeniable influence on the accessibility of health care was not taken into consideration. 

## 3. Results

Hierarchical cluster analysis of average breast cancer mortality identified four groups of countries with similar mortality rates and trends (Table 1). Group 1 included countries with low mortality, which did not change during the analyzed period (about 30%) (constantly low, CL). Countries in Group 2 remained in second place, presenting a declining trend (from 35 to 30%) (decreasing, DEC), thus getting closer to the values achieved by countries in Group 1. Group 3 is characterized by mortality remaining constantly high (about 36%) (constantly high, CH). Group 4, on the other hand, shows an upward trend, reaching an average breast cancer mortality of almost 37% in 2017 (increasing, INC) (Figure 1).

Mortality due to breast cancer in the analyzed period was the lowest in the countries from Group 1 (CL). They ranked second, in terms of the amount of funds allocated to health care (8.48%). The highest financial outlays (9.16%) were in the countries from Group 2 (DEC), where decreasing mortality was observed. The high mortality in Group 3 (CH) corresponds to the amount of funds allocated to health care—in the analyzed group, it was the lowest (6.34%). Similar values were achieved by the countries from Group 4 (INC) (6.7%) (Figure 2).

The largest number of doctors per 100,000 inhabitants was found in the countries of Group 2 (DEC) (375/100,000). Group 1 (CL), with 349 physicians per 100,000 inhabitants (349/100,000), ranked second. In Group 3 (CH), 3.15 doctors per 1000 inhabitants were available (315/100,000). The lowest number of doctors was available in Group 4 (INC)—only 2.9 doctors per 1000 population (290/100,000) (Figure 3).

For countries in Groups 1–3 (CL, DEC, CH), the situation regarding the number of nurses and midwives employed looked similar to the number of available doctors. Group 2 had the highest number of qualified personnel per thousand inhabitants (1045/100,000), followed by Group 1 (CL) (891/100,000) and Group 3 (CH) (576/100,000). However, in the countries from Group 4 (INC), the number of nurses and midwives available per 1000 inhabitants did not reflect the availability of physicians in those countries and ranked third (697/100,000) (Figure 4). 

The number of available magnetic resonance imaging units per 100,000 inhabitants was the highest in group 1 (CL)—2.2/100,000; then, in Group 2 (DEC)—1.51/100,000; in Group 3 (CH)—0.85/100,000. In Group 4 (INC), there were 1.13/100,000 magnetic resonance imaging units (Figure 5). 

The situation was different for the number of computer tomography scanners (CT scanners). The highest number of them per 100,000 inhabitants was in countries in Group 2 (DEC) (2.42/100,000), while slightly fewer were in Group 1 (CL) (2.4/100,000). Countries in Group 3 (CH) had medium CTs availability (2.25/100,000). The fewest CTs were found in countries from Group 4 (INC) (1.98/100,000) (Figure 6).

The highest ratio of mammography examinations performed was observed in countries in Group 1 (CL) (69.5%) and Group 2 (DEC) (69.69%). In contrast, in countries with constantly high and increasing breast cancer mortality, far fewer tests were performed—in Group 3 (CH) (55.85%) and Group 4 (INC) (55.14%) (Figure 7). 

The highest number of beds was available in hospitals in countries with the highest mortality rates. Countries in Group 3 (CH) had as many as 634.45 beds per 100,000 inhabitants. This was followed by countries in Group 4 (INC) with rising rates (523.26/100,000). As many as 467.55 beds per 100,000 inhabitants were available in countries in Group 2 (DEC), and the fewest were in Group 1 (CL) (396.93/100,000) (Figure 8). 

## 4. Discussion

Lalonde’s health field concept states that four factors have an impact on the health of a population—human biology, environment, lifestyle, and health care organization. This study focused on the fourth factor, in the context of breast cancer mortality. The stated aim of the work was achieved through the analysis of health care accessibility and breast cancer mortality in European countries. 

The analysis included 33 European countries with different economic levels, which were categorized into four groups of maximum similarity between countries, in terms of breast cancer mortality rate (Table 1). This allowed individual countries to be allocated to groups with constantly low, decreasing, constantly high, and increasing breast cancer mortality (Figure 1). 

The results of the analysis show that countries with lower mortality (CL and DEC) allocated about twice as much funding as countries where breast cancer mortality was a more common problem (CH and INC) (Figure 2). Cross-sectional studies confirm that large health care expenditures are associated with lower cancer mortality. However, studies of the periods up to and after 2011 in the United States have shown that this is not a linear relationship [13]. The group with the highest investment in both financial and human resources ranked second, in terms of breast cancer mortality rates. Therefore, it can be carefully inferred that countries in Group 2 (DEC) focused on the prevention and early detection of breast cancer by properly allocated investments. Since 2013, the countries of Group 2 (DEC) have reduced the above-mentioned indicators and approached the breast cancer mortality rates of Group 1 (CL) (about 30%). 

The analysis shows that the presence and number of medical personnel is important, in terms of breast cancer mortality. The highest number of available physicians, nurses, and midwives was in the DEC countries, followed by the countries with constantly low breast cancer mortality (Figure 3 and Figure 4). The availability of qualified medical staff was lower in the CH and INC countries. Medical personnel has a significant impact on two factors of Lalonde’s health field concept—health care organization and lifestyle through education. 

The largest, though not similarly large, number of magnetic resonance imaging units was in the CL and DEC countries. In countries where breast cancer mortality is a more common problem, there were fewer units (Figure 5). In contrast, the number of computer tomography scanners is at a similar level in all identified groups. It is worth noting that a higher number of diagnostic equipment is associated with higher spending on the health care system. In addition, the higher number of magnetic resonance imaging units in CL and DEC countries is associated with better accessibility and higher frequency of preventive mammography examinations performed. Despite the lower cost of computer tomography scanning, it is not as detailed as a mammography examination. It is not widely applied to detect breast cancer, although it is useful for diagnosis at the staging stage [14]. 

The largest number and a similar amount of mammography examinations were performed in countries with lower mortality (CL and DEC), while about 15% fewer were performed in countries where breast cancer mortality is more common (Figure 7). Systematic reviews with meta-analysis indicate that participation in mammography screening reduces the risk of death from breast cancer among participants aged 50–69 by up to 33% [15,16]. 

It was observed that countries with low mortality rates had relatively fewer hospital beds per capita (Figure 8). In the 2000s, the role of hospital treatment began to change. High maintenance costs in hospitals made it necessary to reduce the number of hospital beds. To find alternatives to long-term hospital care, emphasis was placed on diagnostics and outpatient treatment [17]. The well-developed medical infrastructure allows for focusing on preventive measures, rather than treatment, thus reducing the need for hospital beds. This trend is also related to the change in the perception of the patient’s role in the treatment process, from passive to active. Outpatient treatment provides greater opportunities for physicians to motivate the patient to therapeutic cooperation and change their attitude towards the disease to one that promotes recovery [18,19]. 

The presented results indicate that high financial expenditure on health care, numerous medical personnel, wide availability of specialized diagnostic equipment, and the prevalence of mammography performers translate into lower mortality rates due to breast cancer. 

The global increase in mortality due to breast cancer contributed to the WHO declaring it the most commonly diagnosed oncological disease, both in the world and in Europe. Lung cancer was previously recognized as the most frequent malignant neoplasm. As of 2020, breast cancer has begun to surpass it in statistics. Poland is characterized by an upward trend in the incidence of breast cancer, which is also observed in other European countries. Over the past 10 years, the total number of breast cancers has increased by almost 100%. However, a disturbing element that distinguishes Poland from other European countries is the rate of decline in mortality. Breast cancer mortality in Poland in the years 1990–2010 decreased by 14.9%, while in Great Britain it decreased by 40%, in Sweden by 30%, and in Czechia and Slovenia—by almost 25% [20]. These data suggest a lower effectiveness of oncological treatment in Poland, compared to other European countries. The comparison of 5-year survival rates of patients diagnosed with breast cancer also confirms this. In patients diagnosed with breast cancer between the years 2005–2009, the 5-year survival rates in Poland were 74.7%. During the same period, rates in Western and South-Western European countries, Slovenia, and Denmark exceeded 80%, whereas in France, Finland, Norway, and Sweden, they exceeded 87% [21]. On the other hand, countries in the groups with constantly low or decreasing breast cancer mortality rates are those where population-based screening was introduced quite early—Sweden (1986), Finland (1987), Great Britain (1988), and the Netherlands (1989) [22]. It should be noted that, in Poland, a population-based breast cancer screening program was introduced in 2006. 

In Poland, there is a breast cancer screening program addressed to women aged 50–69 who meet one of the following criteria: (1) have not had a mammogram in the last two years, (2) have received a written indication to re-mammography after 12 months (within the breast cancer prevention program), due to the burden of the following risk factors (breast cancer among family members (mother, sister, daughter); mutation in the BRCA1 or BRCA2 genes; no previous malignant breast cancer). Despite the widespread availability of preventive examinations and numerous social campaigns to increase knowledge and awareness of breast cancer, attendance to the program is still at a low level—35.33%, with a downward trend [23]. At the beginning of 2021, the World Health Organization (WHO) launched the Global Breast Cancer Initiative, which indicated that the degree of universal health coverage in a country, dedicated funding for the presence of breast cancer early detection programs or guidelines, breast cancer screening programs, a breast cancer referral pathway, having cervical cancer early detection programs or guidelines, and a number of public cancer centers per 10,000 patients with cancer are significantly associated with the breast cancer mortality rate [24]. In contrast to Polish reality, mammography coverage in the countries where the mortality rate had declined in previous years was as follows: over 80% in Denmark and Spain, over 75% in the Netherlands, over 70% in Ireland and Norway, almost 70% in Malta, and about 60% in Czechia, Belgium, and Croatia [4]. 

An increase in breast cancer mortality in Poland can be observed, despite a recent increase in health care funding, which corresponds with the findings presented in the Lalonde report. It has been pointed out that, despite putting most of the social effort and direct expenditures on the organization of health care, the causes of disease incidence and mortality are mainly found in human biology, environment, and lifestyle [1]. Breast cancer mortality is a complex problem that may be influenced by other factors. The presented analysis results and Lalonde’s findings are not the only ones that indicate this. 

The question of the contribution of health care to population health has been the subject of many publications. In 1976, a British physician and scientist, Thomas McKeown, observed that population growth could be attributed to a decline in mortality from infectious diseases, primarily due to better nutrition, later also to better hygiene, and only marginally and later to medical interventions, such as antibiotics and vaccines [25,26]. However, the time when Thomas McKeown undertook his observation was characterized by underdeveloped medicine. This decade also saw the development of the concept of ‘‘avoidable’’ mortality, i.e., premature deaths from selected causes that are considered preventable with appropriate treatment or that could be prevented through adequate health care services. The authors of this concept created a list of diseases that could be treated and/or the appropriate preventive care applied to defer or gradually avoid them. The list of these diseases included breast cancer, among others [26]. The assumptions of the concept of ‘‘avoidable’’ mortality have not only not lost their validity over the past 4 decades, but have even set the course for the development of treatment, diagnosis, rehabilitation, and prevention. Again, it should be emphasized that the country where the above theories were developed, and which took part in shaping the standard of breast cancer prevention, already introduced an effective screening program in the late 1980s. In the presented analysis, the UK is identified as a DEC country, which confirms the validity of long-term and systematic prevention programs and effective education. On the other hand, in Poland, due to the country’s history and the political system that prevailed until 1990, prevention and health promotion activities lag about 20 years behind the statistically “leading” countries. This may also be one of the factors translating into health policy and, consequently, the awareness and early detection of cancer in the Polish population and the Eastern Bloc countries.

The search for factors that influence the increase in breast cancer mortality is extremely important, as their identification will enable planning activities aimed at reducing them. Currently, the probable factors influencing the increase in mortality due to cancer are civilization diseases, extended life expectancy, factors related to females and fertility (as well as the women’s level of awareness and knowledge of preventive measures), habits related to health maintenance behaviors, participation in preventive examinations, and access to medical care. 

The implementation of preventive programs alone does not guarantee that an adequate number of women will voluntarily and regularly participate in screenings. Public health initiatives to prevent or detect cancer at an early stage should be reinforced with reliable and persuasive communication. Women should be aware of the importance of screening, as well as encouraged to repeat the examination at the appropriate time [27]. 

## 5. Conclusions

The results indicating that the higher the financial expenditure on health care and the better health care accessibility, the lower the mortality rates of breast cancer, should not come as a surprise. However, these are factors influenced by governments. Lalonde’s health field concept indicates that health care is neither the only, nor the main, factor that has an impact on the health of a population. The available literature on breast cancer mortality highlights a number of nonmedical factors that influence cancer awareness and early detection. The authors would like to emphasize the need of influencing lifestyle factors and direct cancer risk. However, it is also worth noting that modeling health behavior is a long-term and often difficult process, due to, among others, the cultural, social, and economic conditions of an individual. For this reason, it is important to undertake interdisciplinary actions, combining medical and nonmedical (psychology, sociology, etc.) knowledge, in the process of influencing health behavior. 

## 6. Strengths and Limitations of the Study 

The presented results of the analysis have been limited by the data available to Eurostat. The authors identified systemic factors that could affect breast cancer mortality, but the available data does not exhaust the topic. It is also worth noting that the analyzed data comes from the period 2011–2017, a time before the COVID-19 pandemic. This situation certainly had an impact on access to health care and, consequently, on breast cancer mortality. 

Nevertheless, basing the results on reliable and irrefutable data from Eurostat should be considered a strength of the article. In addition, it should be emphasized that the comparison of the availability of hospital infrastructure and medical staff with the identified groups of countries with similar breast cancer mortality rates and trends may facilitate a discussion regarding the factors influencing the prevention of breast cancer.

## Figures and Tables

**Figure 1 ijerph-19-13605-f001:**
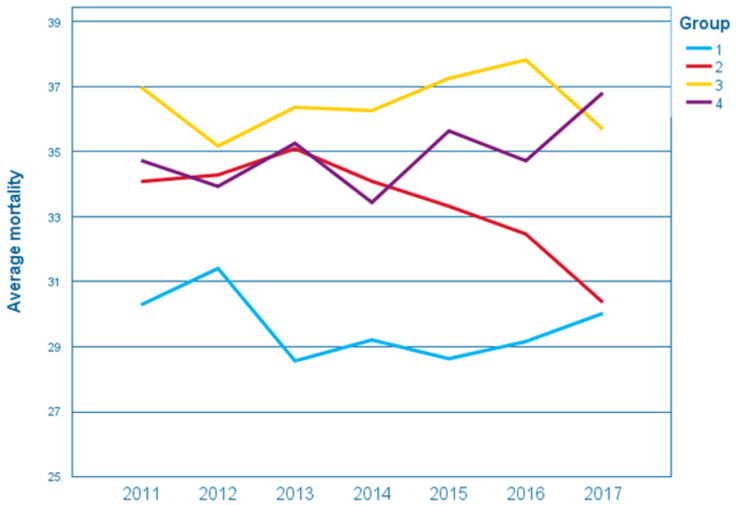
Change of mortality due to breast cancer in the groups of countries.

**Figure 2 ijerph-19-13605-f002:**
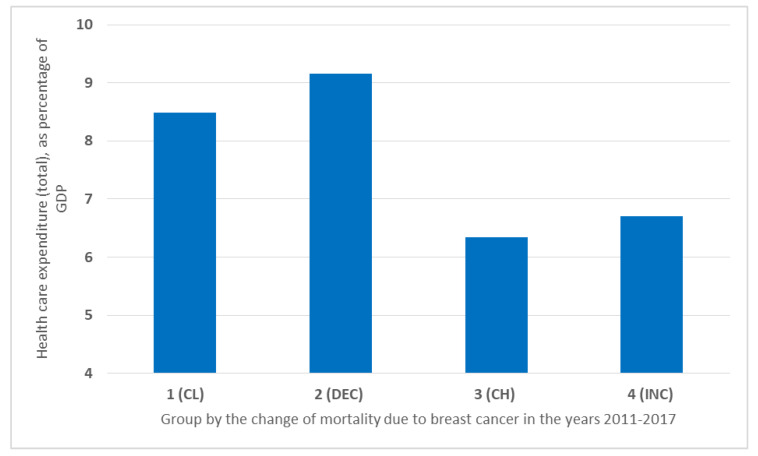
Health care expenditure by all financing agents (total), as a percentage of GDP, in the years 2011–2017.

**Figure 3 ijerph-19-13605-f003:**
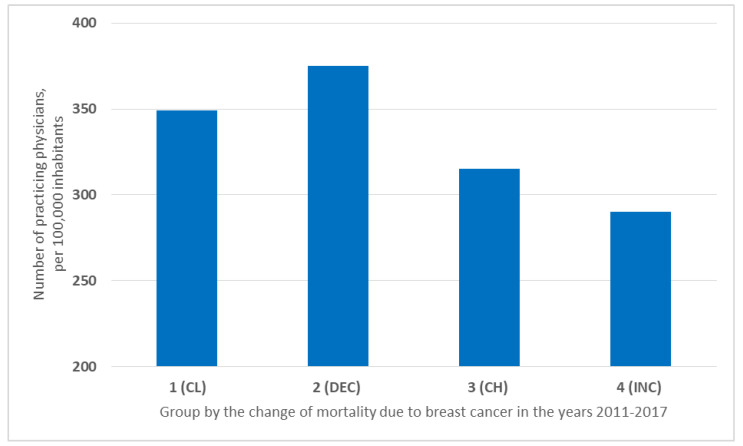
Number of practicing physicians in the group of countries in the years 2011–2017 per 100,000 inhabitants.

**Figure 4 ijerph-19-13605-f004:**
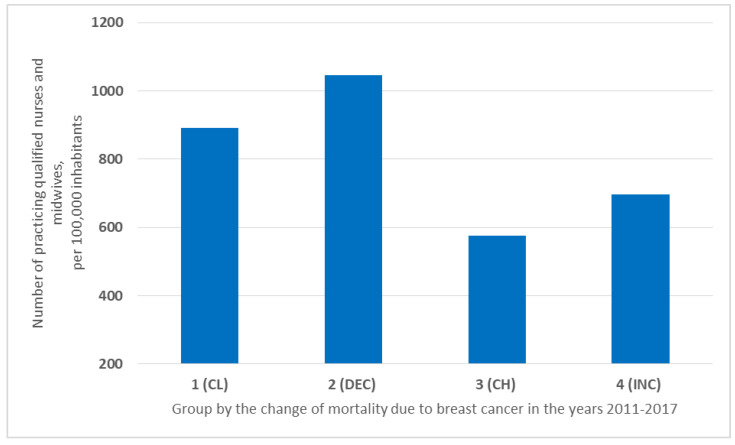
Number of practicing qualified nurses and midwives in the groups of countries in the years 2011–2017 per 100,000 inhabitants.

**Figure 5 ijerph-19-13605-f005:**
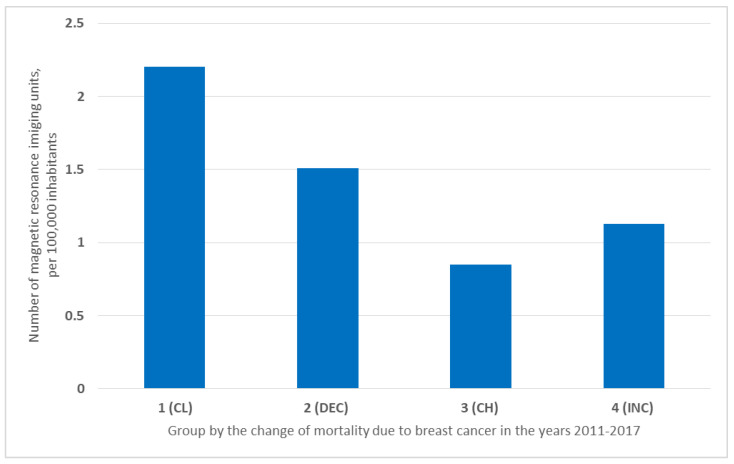
Number of magnetic resonance imaging units in the group of countries in the years 2011–2017 per 100,000 inhabitants.

**Figure 6 ijerph-19-13605-f006:**
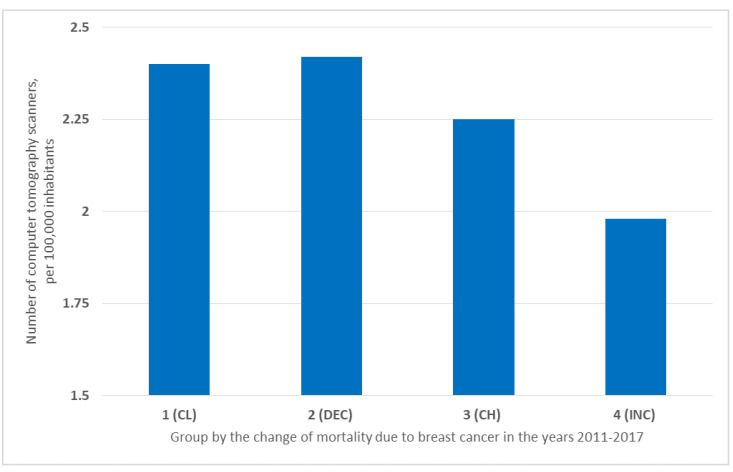
Number of computer tomography scanners in the group of countries in the years 2011–2017 per 100,000 inhabitants.

**Figure 7 ijerph-19-13605-f007:**
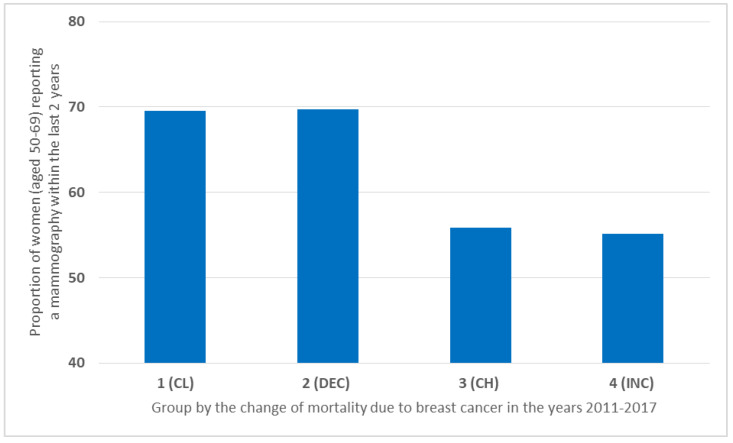
The proportion of women (aged 50–69) reporting mammography in the past 2 years.

**Figure 8 ijerph-19-13605-f008:**
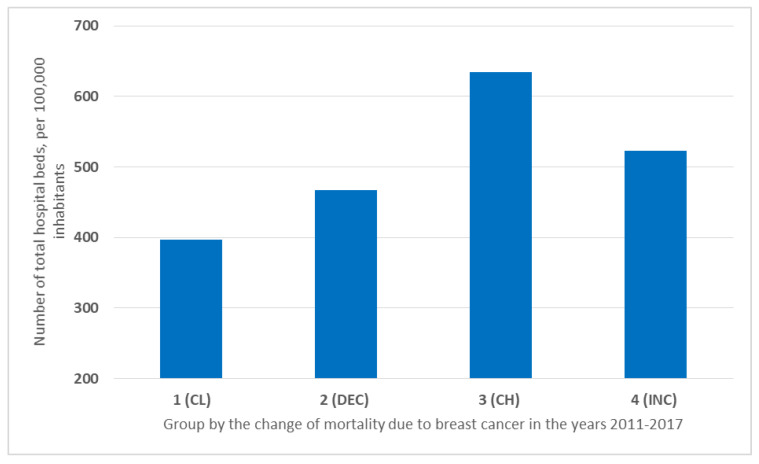
Number of hospital beds in the group countries in the years 2011–2017 per 100,000 inhabitants.

**Table 1 ijerph-19-13605-t001:** Classifying the countries by the change of mortality due to breast cancer in the years 2011–2017.

Group	Countries			
1	Estonia	Finland	Italy	Portugal
2	Austria	Belgium	Bulgaria	Croatia
	Czechia	Denmark	France	Germany
	Iceland	Ireland	Liechtenstein	Lithuania
	Malta	Netherlands	Norway	Spain
	Sweden	Switzerland	United Kingdom	
3	Hungary	Latvia		
4	Cyprus	Greece	Luxembourg	Poland
	Romania	Serbia	Slovakia	Slovenia

## Data Availability

All used data were from Eurostat database and are publicly available at https://ec.europa.eu/eurostat/data/database (accessed on 20 January 2021).

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
