# Peer review of "Health Care Accessibility and Breast Cancer Mortality in Europe"

_ijerph, 2022, doi:10.3390/ijerph192013605_

Round 1

Reviewer 1 Report

1. In Introduction (line 58-59) authors are writing that USG sensitivity and specificity is lower compared to MMG. While this statement is true for specificity, in terms of sensitivity, several studies have shown that the sensitivity of USG is higher compared to MMG; e.g.: 

a) Pereira RO, Luz LAD, Chagas DC, Amorim JR, Nery-Júnior EJ, Alves ACBR, Abreu-Neto FT, Oliveira MDCB, Silva DRC, Soares-Júnior JM, Silva BBD. Evaluation of the accuracy of mammography, ultrasound and magnetic resonance imaging in suspect breast lesions. Clinics (Sao Paulo). 2020;75:e1805.

b) Liu H, Zhan H, Sun D, Zhang Y. Comparison of BSGI, MRI, mammography, and ultrasound for the diagnosis of breast lesions and their correlations with specific molecular subtypes in Chinese women. BMC Med Imaging. 2020 Aug 15;20(1):98.

etc.

2. In Results - I recommend to put the text before the Figure and graph. (swap the text and graph + figure)

3. In Results - most important comment from my side: It is necessary to compare the results statistically and determine the level of significance of the difference in the evaluated parameters between the individual mortality groups (e.g. T-test). This is a necessary step for creating a discussion and formulating the conclusions of the study.

4. The number of performed mastectomies - this data can be misleading. The text does not state the reason for the mastectomy - whether it was an advanced tumor, a condition after neoadjuvant treatment, or just the patient's wish, regardless of the staging of the disease. If the authors have this information, it would be appropriate to include it in the text. Otherwise, recommending mastectomy as a type of surgery to reduce mortality may be misleading and inconsistent with current guidelines (e.g. NCCN). Without supplementing the relevant data, it is better to omit the assessment of this parameter.

Author Response

Dear Reviewer,

We would like to thank you for all suggestions and comments to our article. We appreaciate your professional opinion. Please find attached a detailed answer to each of the comments.

If there are any further concerns or questions on the manuscript content, please let us know.

Sincerely,

Authors 

Reviewer 2 Report

1- It's not clear in the abstract and methodology that the study design and statistical analysis methods

2- In the Introduction, please highlight the rationale and novelty of this study, and here is two related publication that may be used as well on Breast cancer screening and you may found more ;

1- https://doi.org/10.3390/ijerph18010263 

2- Please check if there is a significant difference in terms of mortality of Breast cancer between the categories and within the categories (Between countries ) as well and include that in the analysis, results, and discussion.

3- at the end of the discussion please highlight the strength and the limitations of this study.

4- Please rewrite the conclusions in line with your findings. 

Author Response

(The authors gave the same response as above.)

Reviewer 3 Report

In the present review article Agata Ciuba and colleagues presented data about the “Health care accessibility and breast cancer mortality in Europe”. Current review covers data analysis regarding the subject. The topic and information of this review is somewhat important and relevant considering the elevated number of the affected population. This review is acceptable with the following minor comments:

1.      Abstract: Results- The sentence is too lengthy and not clear, it should be rephrased.

2.      All analysis is presented in histograms, they could have presented in better way, all of them are looking kind of incomplete.

3.      Throughout the review the sentences are too long, especially in discussion part, it makes hard to read and somewhat making the review less engaging.  Modification would likely improve the readability and impact of the manuscript.

Author Response

Dear Reviewer,

We would like to thank you for all suggestions and comments to our article. We appreaciate all your professional opinions. Below you can find a detailed answer to each of the comments. We hope that the changes we introduced to the article will meet your expectations.

If there are any further concerns or questions on the manuscript content, please let us know.

Sincerely,

Authors

Round 2

Reviewer 2 Report

-  Introduction

Many sentences don’t have references, for example, Line 54, Line 64, Line 75, and line 81. Please give more up-to-date references.

The problem statement is mainly about breast cancer mortality and health care accessibility in Europe. However, the introduction doesn’t talk about these points. Likewise, the results are about the financial expenditure on health care, the number of medical personnel, and diagnostic equipment. Nonetheless, the introduction doesn’t talk about these points. Please rewrite the introduction

-        Results

Please improve the presentation of the results in table 1 and include the mortality rate for each group of the countries

-        Discussion

Please support your argument with relevant references

Results are not adequately discussed in the discussion

The discussion is confusing, it is not written well, you need to reflect on your work and evaluate the significance of the data obtained based on your experience, and it should include five main points: (1) a summary of the purpose/goals of the research, (2) theoretical implications of the findings, (3) practical implications of the findings, and (4) strengths, limitations, and recommendations for future studies.

- Conclusions

Most needing reflection is the theoretical and practical implications.

Author Response

Dear Reviewer,

We would like to thank you for all your professional suggestions and comments to our article. They were all very much appreaciated. We hope that the introduction of all changes will make our manuscript ready for publication.

Sincerely,

Authors
